# Laser Fluorescence and Extinction Methods for Measuring the Flow and Composition of Milk in a Milking Machine

**Vladimir V. Kirsanov** [1], **Alexey V. Shkirin** [2,*], **Dmitriy Yu. Pavkin** [1], **Dmitry N. Ignatenko** [2], **Georgy L. Danielyan** [2], **Artyom R. Khakimov** [1] **and Nikolai F. Bunkin** [2,3]

[1] Federal State Budgetary Scientific Institution "Federal Scientific Agroengineering Center VIM", 1st Institutsky proezd 5, 109428 Moscow, Russia; kirvv2014@mail.ru (V.V.K.); dimqaqa@mail.ru (D.Y.P.); arty.hv@gmail.com (A.R.K.)

[2] Prokhorov General Physics Institute of the Russian Academy of Sciences, Vavilova St. 38, 119991 Moscow, Russia; dmitriyek13104@yandex.ru (D.N.I.); gldan@nsc.gpi.ru (G.L.D.); nbunkin@mail.ru (N.F.B.)

[3] Physics Department, Bauman Moscow State Technical University, 2nd Baumanskaya 5, 105005 Moscow, Russia

\* Correspondence: avshkirin@mephi.ru

**Abstract:** Automation of milking systems is linked to accurate measurement of fluctuations in milk flow during milking. To assess the fluctuations of the milk flow, the formation and movement of milk portions in the milking machine-milk pipeline system was studied. By considering the movement of a milk plug along the milk pipeline, a hydraulic model of the formation of a critical volume of milk in the milking machine manifold was compiled. In practice, the most expedient way of determining milk flow parameters may be to measure the laser fluorescent and extinction responses of moving air-milk mixture. We have implemented a new laser sensing method for measuring the flow rate and composition of milk on the basis of counting the optical response pulses received from moving dispersed components by a CCD array or a randomized fiber optic bundle. Using the developed laser sensors, the theoretical model of milk flow was tested.

**Keywords:** laser sensors; milking machines; milk flow fluctuations; milk composition

## 1. Introduction

The study of milk flow fluctuations during milking is of scientific and practical interest in order to accurately measure them for automating the control of milking machines operation modes. In addition, an essential physiological aspect is the control of the end of the milking process in order to exclude overexposure of the teat cups and the harmful effect of vacuum pressure on the animal. At the same time, it is important to minimize the duration of determining the parameters of milk flow, at which the timely switching off and removal of the milking machine should be performed.

Optical methods for monitoring flow inhomogeneities and composition of multiphase liquid media are promising, since they have high sensitivity and speed, and can also carry out non-contact and non-destructive diagnostics [1–14]. Since it is recommended to use devices for measuring milk flow that do not cause a significant pressure drop in the milk hose, optical diagnostic methods can improve the efficiency of the farm equipment. Among them, fluorescence methods should be especially noted as having high selectivity [10–12], as well as light scattering methods [4–8] and methods for measuring optical extinction [13,14] as completely relevant for dispersed media, which represent a random collection of small particles. In the case of milk, these are micelles of caseins and fat globules.

Due to the inertia of chamber flow meters, direct flow meters are increasingly used, for example, the FFS30 infrared milk sensor from SCR (Netanya, Israel) [15] or meters with contact electrodes [16,17]. The simultaneous measurements of milk yield and the

component composition of milk (fat, protein, lactose, etc.) are necessary for determining the productivity and quality of animal milk [18,19]. At the present time, there is no system that provides simultaneous measurement of both the flow rate and the composition of milk in one device. Most modern farms have only a flow meter installed in the milking machine, while the milk quality analyzer is a separate sample analyzer or samples are sent to specialized laboratories for quality analysis. Methods using NIR analyzers with more advanced data processing algorithms are being developed and improved [20–22]. So far, only one system is known that provides a simultaneous analysis of the quality and measurement of milk yield, including a combination of rather complicated devices in operation [23,24]. Thus, the current trend in milking automation is to increase the functionality of milk meters by combining measurements of quantitative (flow rate, milk yield) and qualitative characteristics of milk flow (fat, protein, lactose, etc.), which are recommended to be carried out using the same physical principles, mainly optical spectrometry [25].

It should be borne in mind that since milk contains dissolved air and an ionic component, long-lived ion-stabilized nanobubbles must be present in it [26]. In addition, shear stresses in air-saturated milk lead to the formation of both microbubbles and macrobubbles during transportation in a milk pipeline [27–31]. Thus, it turns out that the milk moving in the pipeline of a milking machine is actually a milk-air mixture. Moreover, the operation of the milking machine in the mode of milk plugs should be particularly considered, when the milk is separated by continuous air gaps in the milk pipeline.

The complexity of the movement of milk-air mixture in the hose of a milking machine and the time limitation cause certain difficulties in measuring a portion of milk when it moves in the flow. Therefore, it is necessary to define a model for estimating milk flow parameters and study the variances of these parameters.

It is important to note that in the process of express assessment of milk quality directly on the farm, specialists are primarily interested in abnormal deviations of the measured milk components in order to take appropriate preventive measures for feeding or treating animals. A more accurate periodic assessment can be carried out already in biochemical laboratories.

In this work, we investigated the processes of transporting milk flows in milking machines in order to find out the possibility of simultaneous measurement of the flow rate, quantity and component composition of milk.

## 2. Methods

For measuring milk flow, international standard ISO 5707 Milking machine installations—Construction and performance recommends using certified devices that cause a pressure drop of no more than +3 kPa and have an error of no more than +5% [32,33]. In milking robots, it is required to control the rate of milk flow by a quarter, which actualizes the creation of small-sized high-precision sensors. The main criteria for assessing the studied parameters of the milk flow can be attributed to: the speed of movement of the milk portion $V_m$ (m/s) and its length $l$ (m), the filling factor of the free section of the hose with milk $k_f$ (%), the volumetric concentration of air bubbles in 1 $cm^3$ of the milk portion $n_{ab}$, the size of the air bubbles $d_{ab}$ (μm), the particle size of the milk constituents (fat, protein, lactose) $d_p$ (μm), the discreteness of scanning the sections of the milk plug along the length $\Delta l$ (m) and in time $\Delta t$ (s). It is obvious that the full functional of the requirements for hybrid milk flow sensors and milk composition can be written as follows:

$$F_\Sigma = f_1\left(q_e : 1\ldots 6\frac{1}{\min}\right) \wedge f_2(\delta : +5\%) \wedge f_3(\gamma : +3 \text{ kPa}) \wedge [f_4(\sigma_{mc} : \sigma_1 \ldots \sigma_n) \veebar f_5(\mu_{mc} : \mu_1 \ldots \mu_n)], \tag{1}$$

where $f_1 \ldots f_5$ are the corresponding functionals: milk flow rates, error relativity, vacuum pressure stability, sampler, milk component composition (fat, protein, lactose, etc.); $\wedge$, $\veebar$ are logical operators.

To estimate the parameters of milk flow theoretically, we propose a model of the formation and movement of milk portions in the system: milking machine-milk pipeline, considering it as a capacitive flow-through hydraulic system with preliminary accumulation of a certain critical volume of milk $W_{cr}^m$ (Figure 1), which consists of two components: the volume of milk in the collector $W_c^m$ and the volume of milk in the milk hose $W_h^m$, corresponding to the arc length $\Delta l_h$ (Equation (2)). Due to the difference in the flow rates of milk ($I_m$) and air ($q_a$), a milk plug is formed.

$$W_{cr}^m = W_c^m + W_h^m \tag{2}$$

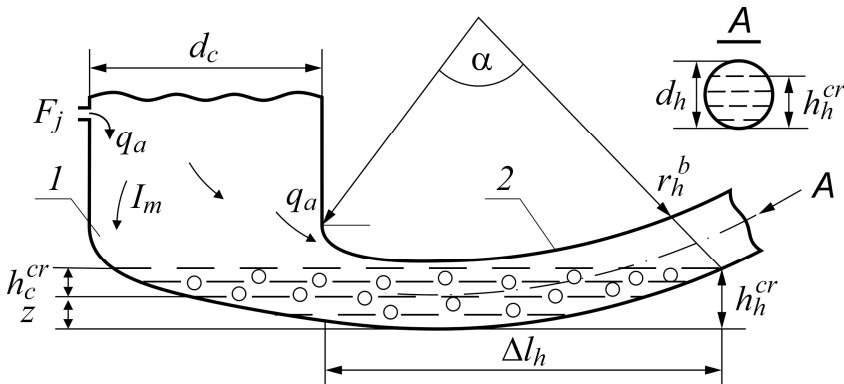

**Figure 1.** Hydraulic model of the formation of a critical volume of milk in the milking machine manifold (teat cups are not shown): (1) collector, (2) milk hose, ($F_j$) air inlet jet, ($q_a$) air flow, ($I_m$) milk flow, ($d_c$) collector diameter, ($d_h$) hose diameter, ($\Delta l_h$) hose arc length, ($h_c^{cr}$) critical milk level in the collector, ($h_h^{cr}$) critical milk level in the hose, ($r_h^b$) hose bend radius, ($z$) difference in height between the lowest point of the collector and the hose, (A) section of the hose.

Under the influence of the resulting pressure drop, the milk plug begins to move along the milk hose into the milk line (Figure 2).

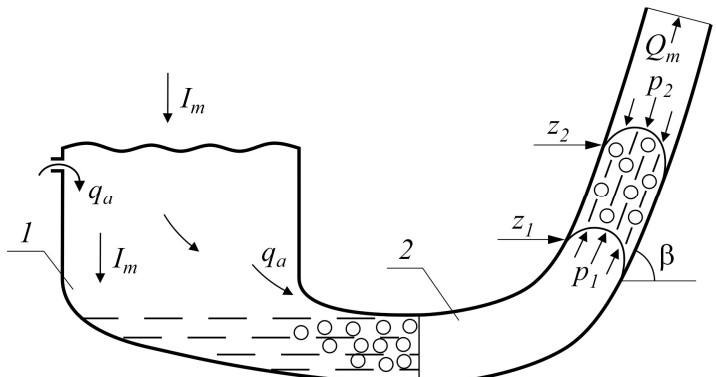

**Figure 2.** Diagram of milk transportation from the milking machine manifold to the milk pipe (not shown): (1) collector, (2) milk hose.

Expression for the volume of milk in the manifold is:

$$W_c^m = \frac{\pi d_c^2 h_c^{cr}}{4} \tag{3}$$

where $d_c$ is the diameter of the milk chamber of the collector (m), $h_c^{cr}$ is the critical milk level in the collector (m). Taking into account the parameters of the hose, we have:

$$W_c^m = \frac{\pi d_c^2 \left(h_h^{cr} - z\right)}{4} \tag{4}$$

where $h_h^{cr}$ is the critical milk level in the hose (m), $z$ is the difference in height between the lowest point of the collector and the hose (m). The critical volume of milk in the hose can be written by analogy as:

$$W_h^m = \frac{\pi d_h^2 \Delta l_h \beta_{cr}}{4} \tag{5}$$

where $\beta_{cr} = h_h^{cr}/d_h$ is the milk hose filling factor, $\Delta l_h = 2\pi r_h^b \alpha$ is the hose arc length, $r_h^b$ is the hose bend radius, $\alpha$ is the center angle corresponding to the hose arc $\Delta l_h$. Thus, expression (5) will be written as follows:

$$W_h^m = \frac{\pi^2 d_h r_h^b \alpha h_h^{cr}}{2} \tag{6}$$

Finally, expression (2) takes the form:

$$W_{cr}^m = \frac{\pi d_c^2 \left(h_h^{cr} - z\right)}{4} + \frac{\pi^2 d_h r_h^b \alpha h_h^{cr}}{2} \tag{7}$$

Next, consider the scheme of the steady-state movement of the milk flow in the hose of the milking machine (Figure 2). For the plug mode, we will use the well-known equation of motion of the milk plug [34], which in the transformed form is written as:

$$\frac{p_1 - p_2}{\rho l} = \alpha \frac{dV}{dt} + g \sin\beta + \frac{\lambda V^2}{2d} \tag{8}$$

where $V$ is the speed of the milk plug (m/s); $p_1$, $p_2$ is the pressure difference on the plug from below and from above, Pa; $l$ is the plug length (m); $\alpha$ is correction for the amount of motion; $\beta$ is the angle of inclination of the hose (degrees); $\rho$ is the density of the milk-air mixture (kg/m³); d is the diameter of the milk hose (m); $\lambda$ is coefficient of hydraulic resistance of the milk hose, g is the acceleration of gravity. The steady-state mode of movement is characterized by the constancy of the milk flow rate in a certain area, where it is advisable to place the milk flow sensor, while the first term on the right side of Equation (8) vanishes and the expression will be rewritten as follows:

$$\frac{p_1 - p_2}{\rho l} = \frac{\lambda V^2}{2d} + g \sin\beta \tag{9}$$

Solving (9) with respect to $V$ leads to the formula:

$$V = \sqrt{\frac{2d(\Delta p - \rho l g \sin\beta)}{\lambda \rho l}} \tag{10}$$

where $\Delta p = p_1 - p_2$ is the pressure difference at the ends of the plug.

Considering that the plug has the shape of a cylinder, its length can be determined through the value of the critical volume of milk $W_{cr}^m$:

$$l = \frac{4 W_{cr}^m}{\pi d_h^2} \tag{11}$$

Taking into account Equation (11), expression (10) for the theoretical speed of the milk plug can be rewritten as:

$$V = \sqrt{\frac{2d\left(\Delta p - \rho\frac{4W_{cr}^m}{\pi d_h^2}gsin\beta\right)}{\lambda\rho\frac{4W_{cr}^m}{\pi d_h^2}}}$$ (12)

Knowing the parameters of the pumping system used in the milking machine, the derived formula allows us to calculate the theoretical speed of the milk plug. The calculated value of the milk plug velocity can be extrapolated to the velocity of the entire pulsating milk-air flow and then the theoretical calculation can be compared with the value of the flow velocity measured in the experimental setup by an optical method.

Obviously, the $W_{cr}^m$ value will vary for different types of milking machines, having a significant effect on the speed and length of milk plugs. At the same time, fluctuations of the density of the milk-air mixture $\rho$ depending on the milk flow rate, contamination of milk-air flows can be, according to various estimates, from 0.5 to ... 0.94 kg/L, having a significant effect on the accuracy of measuring the mass flow of milk [35]. It is quite difficult to theoretically estimate the air saturation indicators of the milk plug due to the strong variation in the indicators of milk flow rate, pressure drop and air flow rate. Therefore, the most appropriate way to solve this problem can be the use of optical spectroscopy methods with measuring the scattering angles of a moving liquid saturated with air bubbles.

Let us consider a schematic diagram of measuring the speed of the movement of a milk plug with steady motion on a segment of hose with a length of $\Delta l$ (Figure 3).

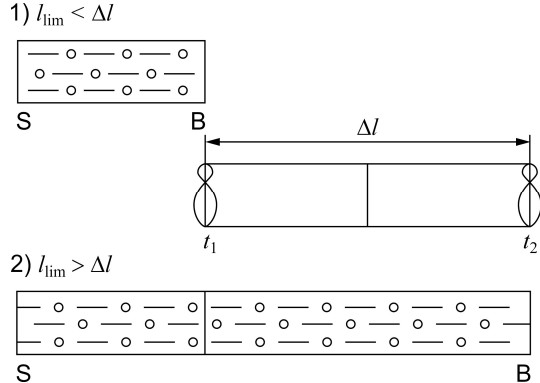

**Figure 3.** Schematic diagram of measuring the characteristics of milk flow: (S) stern part of the milk plug, (B) bow part of the milk plug, ($\Delta l$) hose arc length, ($l_{\lim}$) the limiting length of the milk plug, ($t_1$) and ($t_2$) time moments of entry and exit of the bow of the plug from the hose.

We denote the moments when the bow part of the milk plug passes the initial and final sections, respectively: $t_1$ and $t_2$. With a small assumption, taking into account the low compressibility (extensibility) of the milk plug, we can accept the equality of the velocities of the bow part $V_{pl}^b$ and the stern part $V_{pl}^s$ of the milk plug:

$$V_{pl}^b = V_{pl}^s = V_{pl} = \frac{\Delta l}{t_1 - t_2}$$ (13)

Formula (13) can be used to experimentally determine the flow rate by detecting the moments of entry and exit of a milk plug from the pipe segment at the ends of which

optical detectors are placed. Then the actual parameters of the milk plug (length, volume, mass) and flow rate $q_m$ can be determined by the known expressions:

$$l_{ph} = V_{pl}(t_1^s - t_1^e); \; W_{pl} = \frac{\pi d_h^2 l_{ph}}{4}; \; m_{pl} = \rho W_{pl}; \; q_m = \frac{\pi d_h^2 \rho V_{pl}}{4} \qquad (14)$$

where $t_1^s$, $t_1^e$ are, respectively, the moments of the beginning and the end of the passage of the milk plug of the initial point of the controlled section of the hose.

It should be noted that the method for measuring milk flow using the transit time of the milk plug is known and previously implemented, for example, in [36], while we propose new laser sensing methods based on the discreteness of the milk components.

To measure milk flow parameters, we applied laser fluorescence and extinction methods, which have been implemented in experimental setups with the use of diode lasers, laser beam rasterization, array photodetectors, and multichannel fiber-optic bundles. Information signals were recorded by a digital oscilloscope. Fiber optic probe structures have been optimized for biological fluids, including dairy products, in terms of radiation intensity and signal-to-noise ratio.

## 3. Results

We developed two experimental setups that exhibit the principles of laser fluorescence and extinction for measuring the flow rate and composition of the milk in a milking machine. The measurements of the milk flow velocity were compared with theoretical estimates using the Formula (12).

The influence of air bubbles on the characteristics of milk flow in milking machines was studied experimentally using laser-induced fluorescence method (Figure 4). The setup (Figure 4) employs semiconductor lasers operating at a wavelength of 405 nm and a TCD1304DG CCD linear image sensor (Toshiba, Tokyo, Japan).

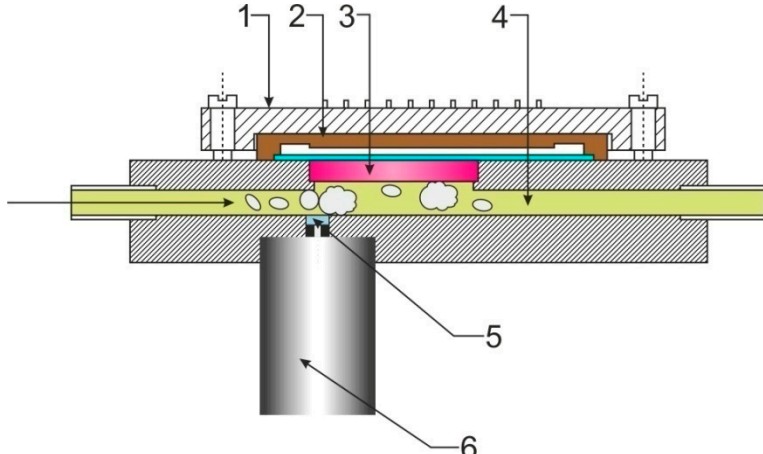

**Figure 4.** Schematic diagram of a laser fluorescent milk flow sensor: (1) holder of CCD array photodetector, (2) SSD array with 3648 pixels, (3) interference band-pass filter for separating fluorescent signal, (4) milk tube; (5) replaceable raster of focused laser radiation, (6) block of lasers emitting in a pulsed modulation mode at a wavelength of 405 nm for the excitation of fluorescence.

To obtain a narrow spatial distribution of the laser beam, we use a raster consisting of a sequence of metal strips deposited on optically transparent glass with equal intervals between them. The raster design is shown in Figure 5.

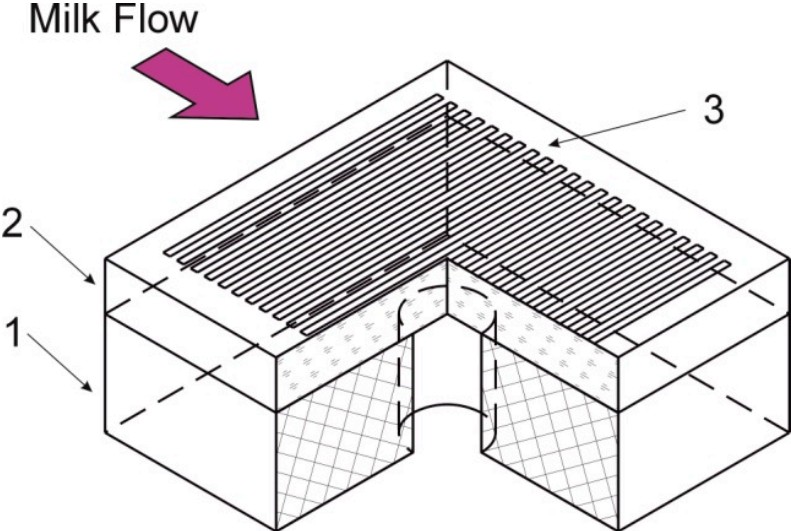

**Figure 5.** Design of a raster: (1) opaque plastic base with a cylindrical aperture, (2) transparent plane-parallel optical glass, (3) metal strips made on the glass surface by Ni-Cr alloy deposition.

We employed two replaceable rasters with a stripe width of 100 and 200 micron. These rasters were attached to an opaque base with a 1 mm hole in the center to narrow the laser beam aperture. The laser beam passed first the hole and then the raster, so that the transverse distribution of the laser beam intensity got rectangular shape, and thus the laser-irradiated area had a narrow width of ~100 microns and sharp boundaries. The raster is left-aligned with a bandpass filter as milk flows from left to right in the diagram shown. This alignment is necessary to record the spatial profile of the fluorescence signal from the portions of milk moving from left to right. The fact is that the fluorescence does not decay immediately and the milk particles excited by the UV laser at the beginning of the CCD array still continue to emit light after passing a distance of ~1000 CCD pixels.

The CCD has 3648 pixels. Actually, not all of the CCD pixels are used. The CCD length is 29 mm and the filter length is 19 mm. Thus, only about 2400 are in use. We deliberately do not use the entire length of the CCD, because on the one hand, the initial pixels may not work distinctly, and on the other hand, the distribution of the fluorescence signal along the CCD has a width of no more than about 2000, so the remaining pixels are not required. Below we show a photograph of the setup (Figure 6), the scheme of which is shown above in Figure 4.

Initial experimental results showed that the fluorescent response propagates and is fixed at distances of up to 20 mm along the length of the milk tube in the sensor (Figure 7). With the passage of air bubbles, a clearly visible change in the slope of the fluorescence intensity distribution along the pipe is observed. The *X*-axis is the pixel numbers of the CCD array. Pixel pitch is 0.008 mm. Pixel range 500 to 3000 corresponds to 20 mm distance along the milk pipeline.

It can be assumed that the small peaks on the fluorescent signal distribution curves (Figure 7) exhibit dispersed milk components, presumably fat globules.

The study of milk-air mixture movement along the milk pipeline by the optical extinction method in the spectral range of 800–850 nm showed that the use of randomized multichannel fiber optic bundles enables the determination of fluctuations in the milk flow. A schematic of the experimental setup is shown in Figure 8. An important feature of the multichannel fiber optic bundles is mixing optical modes, which eliminates the appearance of speckles and heat transfer from the laser heated during operation to the milk pipe.

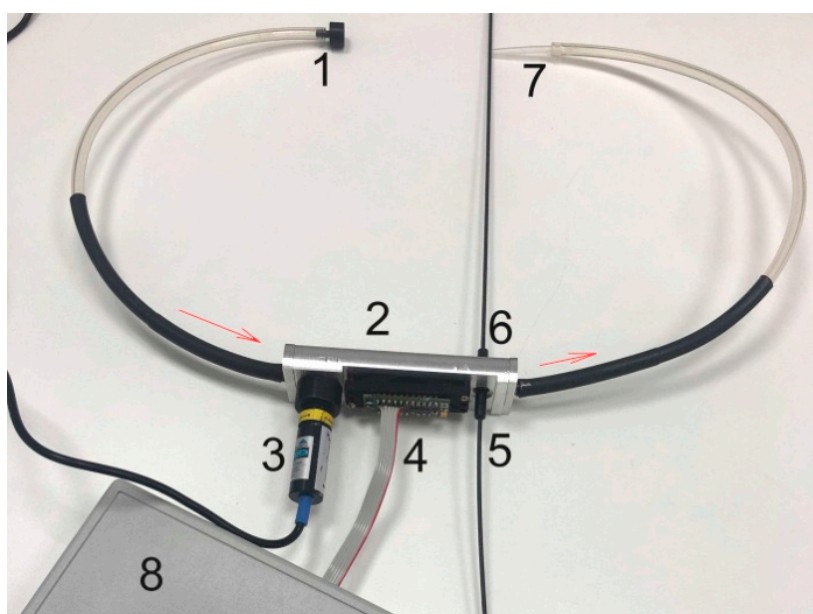

**Figure 6.** Photograph of the laser fluorescent milk flow sensor shown schematically in Figure 4: (1) inlet silicone tube OD-ID 6 × 3 mm² connected to the pump inserted into container with milk, (2) setup holder with 3 mm diameter milk channel, (3) laser (wavelength 405 nm, power 200 mW) with beam raster cutting a 0.1 mm × 2 mm strip from the laser beam, (4) CCD linear array with 3560 pixels (pixel sizes 8 μm × 200 μm), (5) input fiber optic cable connected to LED, (6) output fiber optic cable connected to a photodiode for the test control of relatively large air macrobubbles, (7) outlet silicone tube OD-ID 6 × 3 mm², (8) Electronic unit for converting CCD data into a graph.

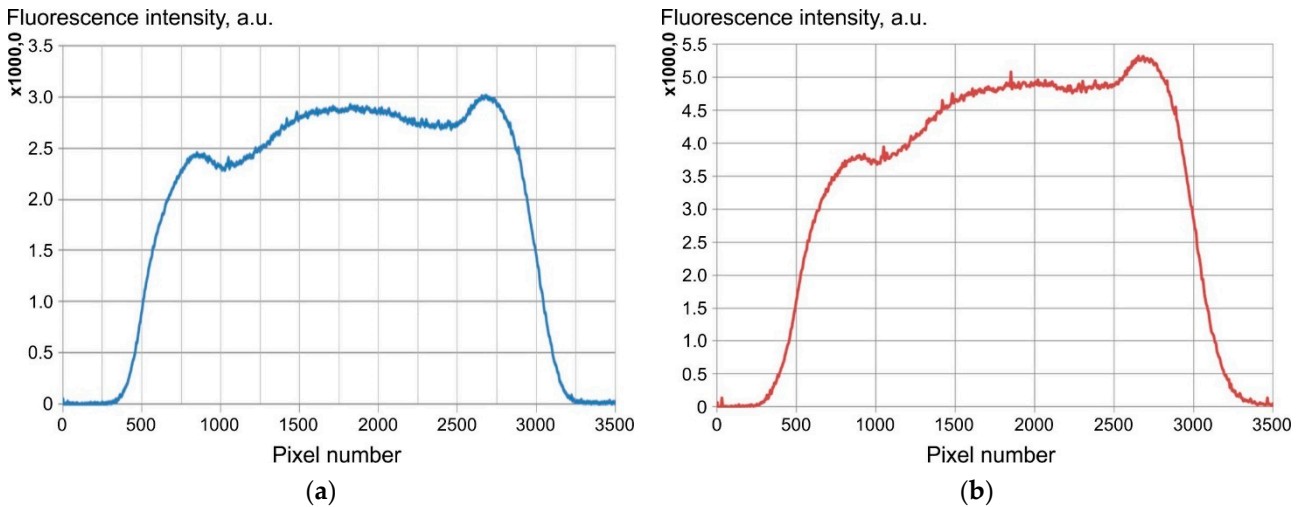

**Figure 7.** Fluorescence intensity distribution along the milk pipeline at the milk flow velocity: (**a**) 0.01 m/s (with bubbles); (**b**) 0.05 m/s (without visible bubbles).

The experimental setup (Figures 8 and 9) provides an imitation of the milking machine operation due to the bubble generator. The milk flow in the milk pipe is not constant, but a succession of air and milk parts of unequal size. The elements 6 and 8 constitute together a fiber optic based optocoupler. Accordingly, the receiver 8 measures the extinction of the intensity of the laser source 6 after passing across the tube in which the milk flows. The optical extinction during the passage of dispersed milk components through the laser beam has the form of short pulses in time. The highest pulses on the oscillograms are obviously given by fat globules as they are the largest among other milk components and

most attenuate the light beam. Since we are dealing with a moving milk-air mixture, the flow of milk is not continuous but fragmented by air gaps (large bubbles). During the passage of a large bubble that approached 8, the receiver 8 stops measuring extinction pulses from small particles of milk (specifically, fat globules) and serves in conjunction with the receiver 7 to detect the number of bubbles passing per unit time. The reference signal coming from the receiver 8 has a double function. First, it is necessary for analyzing the composition of milk by detecting pulses of extinction from dispersed components in the milk plug, primarily fat globules, in order to thus determine the fat content in milk. In addition, assuming that the speed of the milk plug is proportional to the number of pulses per second, the milk flow velocity can be evaluated. Second, it can detect the moment of arrival of large bubbles, which are essentially air gaps in the milk flow. At the same time, the receiver 7 can be used to detect the moment when the bubble passed the distance between 7 and 8; and, in turn, from the bubble transit time, the speed of the entire air-milk flow can be also determined.

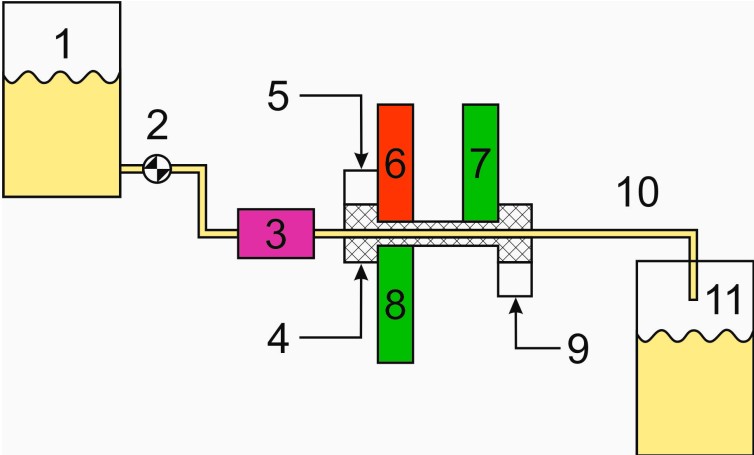

**Figure 8.** Block diagram of the setup for measuring laser extinction with multichannel fiber-optic bundles: (1) milk container, (2) tap, (3) pump, (4) measuring cell, (5) air bubble generator, (6) fiber bundle with laser diode (850 nm, 200 mW), (7) randomized fiber bundle with a photodetector, (8) randomized fiber bundle with a photodetector of the reference signal, (9) photodetector of differential type, (10) model of milk pipe with a cross-sectional diameter of 3 mm, (11) receiving container for milk.

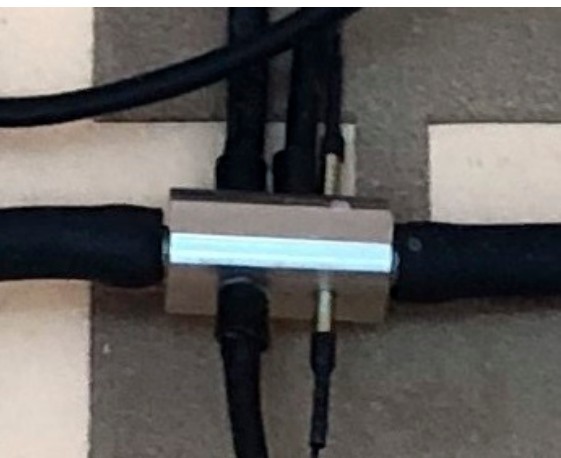

**Figure 9.** Photograph of the laser extinction sensor based on multichannel fiber-optic bundles shown schematically in Figure 8.

Experiments with raw milk using the setup (Figures 8 and 9) showed that the passage of large fat globules can be recorded by reading the optical signal from the fiber optic bundle of the receiver 8 (Figure 10).

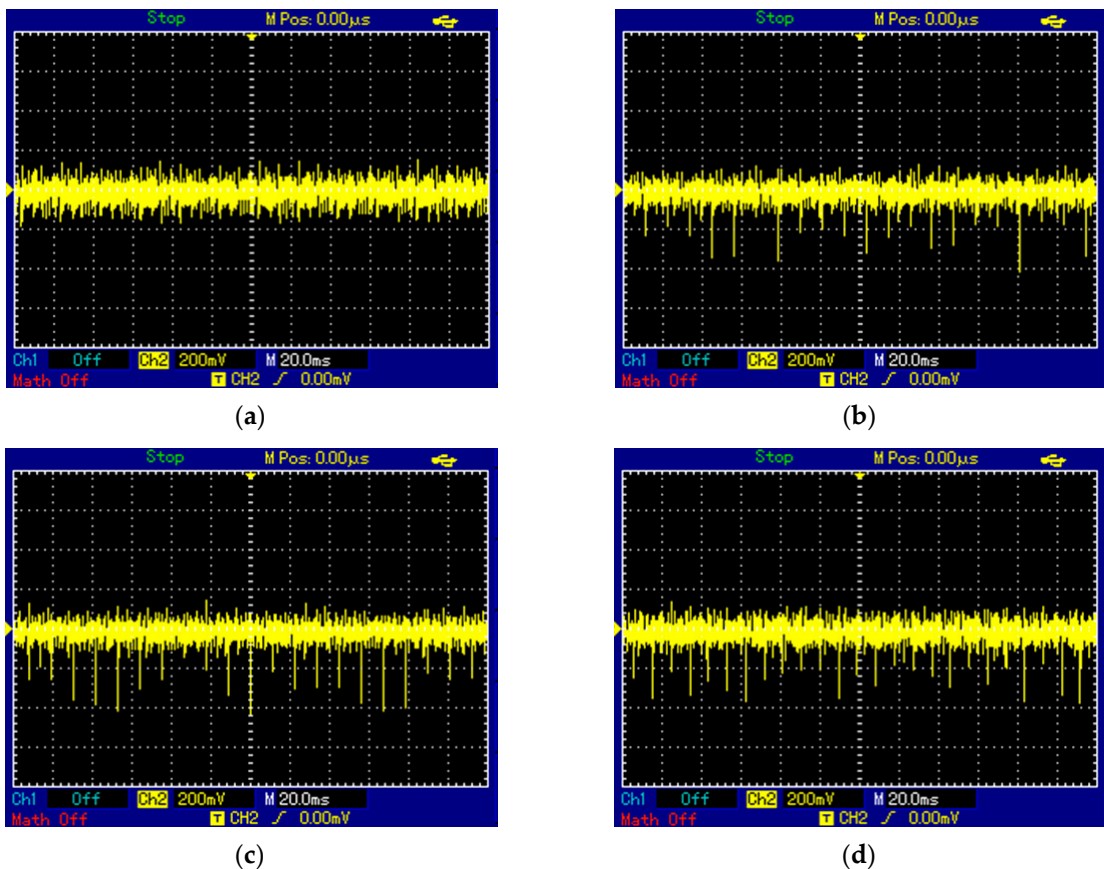

**Figure 10.** Oscillogram screenshots for different milk flow velocities: (**a**) 0 m/s; (**b**) 1.0 m/s; (**c**) 1.3 m/s; (**d**) 1.5 m/s.

The response pulses should be counted at a threshold discrimination of non-informative noise. As can be seen from the oscillograms of recorded signals (Figure 10), small-sized impulse emissions most likely correspond to optical extinction on small fat globules, and sharp peaks in the minus region are mainly associated with the presence of large fat globules. When milk moves along the milk pipe, these minus peaks change the frequency of their appearance.

Based on the recorded waveforms (similar to those in Figure 10), we have plotted a graph (Figure 11) that illustrates a direct correlation between the flow velocity and the frequency of negative pulses $N_p$ in discrete extinction pattern of the flowing milk. Experimental values of $N_p$ were obtained by counting the number of pulses exceeding the threshold noise value, estimated from the oscillograms as the average height of uniform noise ($-100$ mV). Statistical averaging over a set of sequentially recorded signals was performed; mean values of $N_p$ with standard deviations and a linear approximation are displayed in Figure 11.

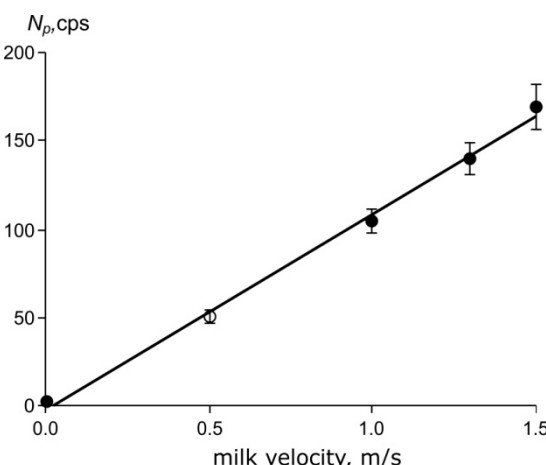

**Figure 11.** Influence of the milk velocity on the counting rate of response pulses $N_p$, measured as the number of counts per second (cps) in the reference signal.

The dependence (Figure 11) can be considered as an experimental test of the outlined milk flow model, since $N_p$ (ordinate axis) was calculated as the average number of pulses per second over several waveform oscillograms measured for each of the given values of flow velocity (abscissa axis), which were set in the pumping system of the experimental setup using the relationship of speed and pressure according to Formula (12). The adjusted values of flow rate were checked by measuring the volume of the milk passed through the sensor during the test using a graduated cylinder.

The approximate linearity of the obtained dependence (Figure 11) indicates the adequacy of the applied model of laminar flow for air-milk mixture. Thus, the measurements of response counting rate can in principle be used to determine both milk flow rate and fat content.

## 4. Discussion

The applicability of laser sensing methods for assessing the flow rate and composition of dairy products in pipelines of milking complexes has been studied experimentally. Our experiments have shown that using the methods of laser-induced fluorescence response and laser extinction together with digital signal processing algorithms, it is possible to control the flow rate and amount of dispersed components of milk, in particular fat globules, in the milk pipeline. The tested technique for the monitoring of milk flow and composition in a pipe with laser/LED illumination in the 800–850 nm spectral region is based on the optical extinction properties of fat globules of about 0.01 . . . 10 µm in size, contained in milk in large quantities. When measuring the fat content in raw milk using the proposed technique, the presence of fat globule agglomerates up to 100 µm in size must be taken into account. It should be noted that exact measurement of milk fat percentage requires a calibration of the sensor.

## 5. Conclusions

The use of the proposed analytical model of milk flow through the pipeline of a milking machine in conjunction with laser sensing methods provides a physical basis for simultaneous measurements of the flow characteristics and composition of milk. The detection of optical response pulses from moving fat globules can serve as a principle for measuring the flow rate and fat content of milk, when milking cows or processing dairy products in technological lines. Such a new type of laser milk sensors can be developed using laser diodes or LED with different wavelengths from UV to near-IR range together with CCD array photodetectors and multichannel fiber bundles.



**Author Contributions:** Conceptualization, V.V.K.; methodology, D.Y.P.; validation, V.V.K.; formal analysis, A.R.K.; investigation, A.R.K., G.L.D. and D.N.I.; data curation, N.F.B.; writing—original draft preparation, D.Y.P., G.L.D.; writing—review and editing, A.V.S., N.F.B.; visualization, D.Y.P.; supervision, A.V.S.; project administration, V.V.K. All authors have read and agreed to the published version of the manuscript.

**Funding:** This work was supported by a grant of the Ministry of Science and Higher Education of the Russian Federation for large scientific projects in priority areas of scientific and technological development (grant number 075-15-2020-774).

**Institutional Review Board Statement:** Not applicable.

**Informed Consent Statement:** Not applicable.

**Data Availability Statement:** Not applicable.

**Acknowledgments:** Not applicable.

**Conflicts of Interest:** The authors declare no conflict of interest.

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
