# Peer review of "Laser Fluorescence and Extinction Methods for Measuring the Flow and Composition of Milk in a Milking Machine"

_photonics, doi:10.3390/photonics8090390_

Round 1

Reviewer 1 Report

In the MS the authors proposed to use the laser fluorescence to measure the flow rate of the milk in the miking machine and through observe the small size pluses in the optical signal to determine the composition of the milk. Although the idea is interesting, more convincing results are needed. With the current content, I can not suggest it to be published in Photonics. In the MS only the experiment (about 2 pages) is related to laser, this occupies very small portion. I think this MS would be more suitable to be published in journals related to fluid or agriculture. A significant revision is required before my further consideration. Please considering the following issues.

More Photonic related content needs to be added. I’m unable to see a clear connection between experiment (page 6-7) and all those theories, formulas in page 3-5, the experiment is supposed to verify the theories. All those formulas should somehow be able to apply in the experiment. In this MS, the experiment seems independent from the rest of sections.

There are few issues with figure 4:

Where is the CCD array photodetector? On the figure you only showed (1) is the holder, but where the photodetector device located?

What is the blue material, in between 2 and 3?

From the figure SSD is not directly scanning the milk flow in the tube, under the SSD you have the pink material which is the band pass filter. This band pass filter is clearly shorter than the SSD, therefore the length of the filter determines the active pixels of SSD. Please indicates how many pixels are actually in use.

Please explain the use of 5, raster? The position of 5,6 are not make sense, why it is right aligned with the band pass filter? Wouldn’t be better if the laser source is placed in the middle of the band pass filter?

As for figure 6, should the position of 7 and 8 be exchanged? 7 should be the reference signal, because there is no milk passing through between 6 and 7.

More detail explanation is needed for results in fig7 and 8.

Where are the results in fig7 coming from? From block 7 in fig6?

How is the reference signal used in this experiment?

How do you define the threshold noise value, and what is the threshold value?

What is the y axis, Np cps?

Author Response

Response to Reviewer 1 Comments

Point 1: In the MS the authors proposed to use the laser fluorescence to measure the flow rate of the milk in the miking machine and through observe the small size pluses in the optical signal to determine the composition of the milk. Although the idea is interesting, more convincing results are needed. With the current content, I can not suggest it to be published in Photonics. In the MS only the experiment (about 2 pages) is related to laser, this occupies very small portion. I think this MS would be more suitable to be published in journals related to fluid or agriculture. A significant revision is required before my further consideration. Please considering the following issues. 

 Response 1: Thank you for carefully reading our manuscript. The fact is that we submitted the manuscript to a special issue "Driving Spectroscopy and Laser Physics toward Biological, Agricultural, and Medical Applications". Since the aim of the special issue is "to collect studies dealing with optical and laser spectroscopy techniques which forward the knowledge of biology and medicine with fundamental as well as new perspective applications of photonics studies for ecology, agriculture, and the food industry," we believe that the material of the manuscript corresponds to the subject of the issue.

Point 2: More Photonic related content needs to be added. I’m unable to see a clear connection between experiment (page 6-7) and all those theories, formulas in page 3-5, the experiment is supposed to verify the theories. All those formulas should somehow be able to apply in the experiment. In this MS, the experiment seems independent from the rest of sections.

Response 2: The need for a theoretical consideration of milk flow in plug mode, arising from the injection of air is justified by the implementation of such an operation mode of milk machine using a bubble generator in our experiments. We have added a more detailed description of the operation principle along with photographs of the optical setups used in the experiment. We have added a description of how the final formula 12 derived in theoretical section is applied in obtaining the experimental dependence in Fig. 8 (Fig. 11 in the revised version). We have also clarified the purpose of the last part of the theory with the formulas 13 and 14 that are used for the experimental determination of flow characteristics through the detection of the plug entrance and exit moments by fiber optic receivers.

Point 3: There are few issues with figure 4:

Where is the CCD array photodetector? On the figure you only showed (1) is the holder, but where the photodetector device located?

Response 3: We have specified the type of CCD that is installed in the experimental setup: Toshiba CCD linear image sensor TCD1304DG. We filled in a special color the section of the CCD array (2) in the drawing.

Point 4: What is the blue material, in between 2 and 3?

Response 4: The blue material is the outer glass of the CCD sensor. We have made appropriate designation in the drawing and indicated it in the caption to the figure.

Point 5: From the figure SSD is not directly scanning the milk flow in the tube, under the SSD you have the pink material which is the band pass filter. This band pass filter is clearly shorter than the SSD, therefore the length of the filter determines the active pixels of SSD. Please indicates how many pixels are actually in use.

Response 5: Yes, actually not all of the CCD pixels are used. The CCD length is 29 mm and the filter length is 19 mm. Thus, of all the 3648 pixels, only about 2400 are in use. We deliberately do not use the entire length of the CCD, because on the one hand, the initial pixels may not work very clearly, and on the other hand, the distribution of the fluorescence signal along the CCD has a width of no more than about 2000, so the remaining pixels are not required.

Point 6: Please explain the use of 5, raster? The position of 5,6 are not make sense, why it is right aligned with the band pass filter? Wouldn’t be better if the laser source is placed in the middle of the band pass filter?

Response 6: Raster is a sequence of 100 or 200 micron wide metal strips deposited on optically transparent glass with equal intervals between them. This structure was attached to an opaque base with a 1 mm hole in the center to narrow the laser beam aperture. The laser beam passed first the hole and then the raster. Such a raster was necessary for the transverse distribution of the laser beam intensity to take on a rectangular shape, and thus the laser-irradiated area had a narrow width of ~ 100 microns and sharp boundaries. We have added a separate figure showing the raster design. We would like to note that the raster is left-aligned with a bandpass filter as milk flows from left to right in the diagram shown. This alignment is necessary to record the spatial profile of the fluorescence signal from the portions of milk moving from left to right. The fact is that the fluorescence does not decay immediately and the milk particles excited by the UV laser at the beginning of the CCD array still continue to emit light after passing a distance of ~ 1000 CCD pixels.

Point 7: As for figure 6, should the position of 7 and 8 be exchanged? 7 should be the reference signal, because there is no milk passing through between 6 and 7.

Response 7: The elements 6 and 8 constitute together a fiber optic based optocoupler. Accordingly, 8 measures the extinction of the intensity of the laser source 6 after passing across the tube in which the milk flows. The optical extinction during the passage of dispersed milk components through the laser beam has the form of short pulses in time. The highest pulses on the oscillograms are obviously given by fat globules as they are the largest among other milk components and most attenuate the light beam. Since we are dealing with a moving milk-air mixture, the flow of milk is not continuous but fragmented by air gaps (large bubbles). During the passage of a large bubble that approached 8, the receiver 8 stops measuring extinction pulses from small particles of milk(specifically, fat globules) and serves in conjunction with 7 to detect the number of bubbles passing per unit time. So, exchanging the position of 7 and 8 does not make sense.

Point 8: More detail explanation is needed for results in fig7 and 8.

Response 8: We have added a more detailed description of the algorithm for receiving and processing the recorded signals and also explanations on the interrelation of the number of extinction pulses per second and the flow rate.

Point 9: Where are the results in fig7 coming from? From block 7 in fig6?

Response 9: The resulting signals on the oscillograms in Fig. 7 (Fig. 10 in the revised version) came from the receiver 8. Receiver 7 is auxiliary and is needed to determine the number of bubbles of large bubbles clogging the entire tube, that is, forming air plug by detecting the time of their movement between 8 and 7.

Point 10: How is the reference signal used in this experiment?

Response 10: The reference signal coming from 8 have a double function. First, it is necessary for analyzing the composition of milk by detecting pulses of extinction from dispersed components in the milk plug, primarily fat globules, in order to thus determine the fat content in milk. In addition, assuming that the speed of the milk plug is proportional to the number of pulses per second, the milk flow velocity can be evaluated. Second, it can detect the moment of arrival of large bubbles, which are essentially air gaps in the milk flow. At the same time, 7 is used to detect the moment when the bubble passed the distance between 7 and 8; and, in turn, from the bubble transit time, the speed of the entire air-milk flow can be also determined.

Point 11: How do you define the threshold noise value, and what is the threshold value?

Response 11: Threshold value is the average height of uninformative uniform noise. (-100 mV).

Point 12: What is the y axis, Np cps?

Response 12: Np is the number of pulses per second. “cps” is a common abbreviation for “counts per second”.

Reviewer 2 Report

The paper has material to be published. However, it needs to be re-structured. you need to separate the methodology from the results. The methodology has to include all the mathematical equations and experimental setup. The experimental technique is not well presented as I can only understand it from the title of the figure.  Results should not include the mathematical analysis or the experimental test rig but should include the output of the model and analysis of these data. At the moment, the paper looks like a summary of a thesis that has been put together in a hurry. The model is not well presented and has not much detail about the proposed equations. Equation 8 needs revising as it is not consistent in units or you could provide derivation of what has been neglected (I assume Sin (B) is multiplied and not added). The same applies to the experimental technique. The results section has not included analysis for the raw data you obtained to support your conclusions. I cannot see also results about measuring the composition apart from fluorescence intensity distribution. You may need to do calibration to be able to measure the composition and compare with other methods such as the common hex-ane:isopropanol solvent extraction method.
I encourage to restructure the paper and re-submit. 

Author Response

Response to Reviewer 2 Comments

Point 1: The paper has material to be published. However, it needs to be re-structured. you need to separate the methodology from the results. The methodology has to include all the mathematical equations and experimental setup. The experimental technique is not well presented as I can only understand it from the title of the figure.  Results should not include the mathematical analysis or the experimental test rig but should include the output of the model and analysis of these data. At the moment, the paper looks like a summary of a thesis that has been put together in a hurry. The model is not well presented and has not much detail about the proposed equations. Equation 8 needs revising as it is not consistent in units or you could provide derivation of what has been neglected (I assume Sin (B) is multiplied and not added). The same applies to the experimental technique. The results section has not included analysis for the raw data you obtained to support your conclusions. I cannot see also results about measuring the composition apart from fluorescence intensity distribution. You may need to do calibration to be able to measure the composition and compare with other methods such as the common hex-ane:isopropanol solvent extraction method.

I encourage to restructure the paper and re-submit.

 Response 1: Thanks for the fair comments. We have made appropriate changes in the text. We have outlined the methodology more clearly. The main purpose of the mathematical derivations in the theoretical part was to obtain the formula 12 for calculating the flow rate from the known parameters of the pumping system. While, the last formulas 13 and 14 describe a simple algorithm for experimentally estimating the speed of a milk plug through optical detection of the plug entrance and exit moments. We have corrected the error in the equation 8 “sin B” is replaced by “g sin B” (g is the acceleration of gravity). We specifically formulated the objectives of the experiments by describing what quantities were measured and how the formulas of the theory were applied. We have also added a more detailed description of the operation principles of the setups used in the experiment and inserted their real photographs. We have explained what we mean by measuring the composition of milk, it is the number of fat globules per unit volume that gives the value of the fat content. We also explained how the presence of bubbles and dispersed components in air-milk mixture can be detected and then used to measure the flow characteristics (firstly, the velocity) by the optical methods that we used. We describe how raw data taken from the oscillograms of the measured signals in Fig. 7 (Fig. 10 in the revised version) were analyzed to plot the graph in Fig. 8 (Fig. 11 in the revised version). The number of pulses per second (ordinate axis) was calculated from them, while the flow velocity values in the pumping system of the experimental setup (ordinate axis) were set using calculations according to formula 12. The linearity of the obtained dependence in Fig. 8 (Fig. 11 in the revised version) indicates the adequacy of the applied model of laminar flow for air-milk mixture.

Round 2

Reviewer 1 Report

I'm happy with authors' response, agree to publish

Author Response

Thank you a lot for your positive evaluation of our revision of the manuscript

Reviewer 2 Report

Much better version and fast turn up. As the paper is mainly developing a new methodology, all the mathematical analysis and system components should be under the methodology title and not results. Results may contain the signal and signal analysis section to prove that the system do work and give results. The conclusions section is very long single statement, which could be simplified. I would also highlight that the composition can be evaluated in principle as you still need to do calibration to be able to identify the composition as a percentage. I understand this is not an easy task and it may need a separate paper.

Author Response

Response to Reviewer 2 Comments

Point 1: Much better version and fast turn up.

Response 1: Thank you very much for your positive evaluation of the manuscript revision and additional valuable comments. We have made appropriate improvements in the new revision of the manuscript that are highlighted with green. Hope that the quality of our manuscript is now acceptable.

Point 2: As the paper is mainly developing a new methodology, all the mathematical analysis and system components should be under the methodology title and not results.

Response 2: We admit that it is more appropriate to place formulas in the methodology section. We have moved the mathematical part there accordingly.

Point 3: Results may contain the signal and signal analysis section to prove that the system do work and give results.

Response 3: The signal analysis was quite simple; we merely calculated the statistical average (arithmetic mean) and standard deviation of the number of pulses exceeding the threshold per unit of time over a set of signals sequentially recorded by a digital oscilloscope. We have mentioned this analysis routine in the newly revised version.

Point 4: The conclusions section is very long single statement, which could be simplified.

Response 4: We have simplified the too long statement of the conclusion by breaking it down into separate aspects.

Point 5: I would also highlight that the composition can be evaluated in principle as you still need to do calibration to be able to identify the composition as a percentage. I understand this is not an easy task and it may need a separate paper.

Response 5: We fully agree that we showed only a fundamental possibility of measuring the milk composition. We highlighted in the conclusion of the new version that the proposed technique provides just a physical basis for evaluating the composition. Exact measurement of the percentage of fat certainly requires a calibration, but this is rather a subject for a further research.
